# Qualitative Transcriptional Signature for Predicting the Pathological Response of Colorectal Cancer to FOLFIRI Therapy

**DOI:** 10.3390/ijms252312771

**Published:** 2024-11-27

**Authors:** Jun He, Mengyao Wang, Dandan Wu, Hao Fu, Xiaopei Shen

**Affiliations:** 1Fujian Key Laboratory of Medical Bioinformatics, Department of Bioinformatics, Institute of Precision Medicine, School of Medical Technology and Engineering, Fujian Medical University, Fuzhou 350122, China; wangmy2022@gmail.com (M.W.); haofu@fjmu.edu.cn (H.F.); 2Key Laboratory Gastrointestinal Cancer (Fujian Medical University), Ministry of Education, Fuzhou 350122, China; 3School of Nursing, Fujian Medical University, Fuzhou 350122, China; dandanw1994@163.com

**Keywords:** metastatic colorectal cancer, FOLFIRI, response, gene signature

## Abstract

FOLFIRI (5-FU, leucovorin, irinotecan) is the first-line chemotherapy for metastatic colorectal cancer (mCRC), but response rates are under 50%. This study aimed to develop a predictive signature for FOLFIRI response in mCRC patients. Firstly, Spearman’s rank correlation and Wilcoxon rank-sum test were used to select chemotherapy response genes and gene pairs, respectively. Then, an optimization procedure was used to determine the final signature. A predictive signature consisting of three gene pairs (3-GPS) was identified. In the training set, 3-GPS achieved an accuracy of 0.94. In a validation set of 60 samples, predicted responders had significantly better progression-free survival than the predicted non-responders (HR = 0.47, *p* = 0.01). A comparable result was observed in an additional validation set of 27 samples (HR = 0.06, *p* = 0.02). The co-expressed genes of the signature were enriched in pathways associated with the immunotherapy response, and they interacted extensively with FOLFIRI-related genes. Notably, the expression of signature genes significantly correlated with various immune cell types, including plasma cells and memory-resting CD4+ T cells. In conclusion, the REO-based signature effectively identifies mCRC patients likely to benefit from FOLFIRI. Furthermore, these signature genes may play a crucial role in the chemotherapy.

## 1. Introduction

Colorectal cancer (CRC) is the third most common cancer worldwide [1]. Approximately 25% of CRC patients are diagnosed with advanced disease at the initial diagnosis, and an additional 35% develop metastases during the course of the disease [2]. Despite recent advances in immunotherapy and targeted therapy, chemotherapy remains the primary treatment option for patients with metastatic colorectal cancer (mCRC) [3]. One of the first-line chemotherapy regimens for mCRC is the combination of 5-FU, leucovorin, and irinotecan (FOLFIRI) [4]. However, the response rate of FOLFIRI for stage III metastatic CRC and advanced CRC are only 56% [5] and 31% [6], respectively. Furthermore, there is still a lack of biomarkers that can reliably predict the likelihood of a patient benefiting from FOLFIRI treatment.

To address this issue, transcriptional signatures have been developed to predict the response of mCRC to FOLFIRI chemotherapy [7,8]. However, these signatures are often derived from quantitative gene expression data, which may be influenced by confounding factors such as batch effects, arising from differences in laboratory conditions and personnel operations [9,10]. Thus, it is crucial to normalize the data. This requires the normalization of a single sample in conjunction with a group of samples, which presents a challenge in the application of gene markers in a general clinical context. In contrast, it has been demonstrated that the relative expression order (REO) of gene pairs in samples is resistant to the effects of batch effects, differing measurement principles across different platforms, the uncertainty of the sampling location in tumor tissues, and partial RNA degradation [11,12,13].

The objective of this study is to construct a reliable qualitative transcriptional signature based on the within-sample REO of gene pairs for the purpose of predicting the response to FOLFIRI treatment in patients with mCRC.

## 2. Results

### 2.1. Development of the Signature for FOLFIRI

The discovery workflow is illustrated in Figure 1. Using the gene expression of 21 mCRC patients from CRC21, 715 FOLFIRI chemotherapy response genes related to tumor lesions size were screened (Spearman’s rank correlation, *p* < 0.05 and |r| > 0.3). Subsequently, 14,646,775 gene pairs comprising at least 1 response gene were identified through the pairwise combination of all genes. Furthermore, based on 47 mCRC patients from CRC47, a total of 72,286 gene pairs whose REO patterns were significantly associated with the response outcomes were selected (Wilcoxon rank-sum test, *p* < 0.01). Based on these gene pairs, a set of three gene pairs with the highest F-score was selected as the predictive signature, designated as 3-GPS (Table 1). The samples were scored in accordance with the number of gene pairs exhibiting the specific REO (Gene i > Gene j) of the 3-GPS. A sample was classified as a responder if the score reached 2. Otherwise, the sample was classified as a non-responder.

### 2.2. Performance of the 3-GPS

In the training set CRC47, 33 of the 36 responding samples and all 11 non-responding samples were correctly classified (sensitivity = 0.92 and specificity = 1.00) by 3-GPS. The AUC and accuracy were 0.95 and 0.94, respectively (Figure 2A). Furthermore, the AUC could reach 0.94 and 1.00 in CRC47-GSE72970 and CRC47-GSE62080, respectively (Figure 2B and Figure 2C).

The 3-GPS was validated in two datasets: CRC60 and CRC27. In the validation dataset CRC60, Kaplan–Meier curves and univariate Cox analysis showed that the patients predicted as responders by 3-GPS exhibited significantly superior PFS compared to those predicted as non-responders (univariate Cox, HR = 0.53, 95% CI = 0.30–0.91, *p* = 1.9 × 10^−2^; Figure 3A). Similar results were also observed in CRC27 (univariate Cox, HR = 0.15, 95% CI = 0.04–0.56, *p* = 1.5 × 10^−3^; Figure 3B) and in the pooled datasets of CRC60 and CRC27 (univariate Cox, HR = 0.45, 95% CI = 0.28–0.72, *p* = 6.6 × 10^−4^; Figure 3C). After adjusting for age, gender, stage, and tumor location, the 3-GPS remained statistically significant in CRC60 (multivariate Cox, HR = 0.43, 95% CI = 0.23–0.83, *p* = 0.01; Figure 3D), CRC27 (multivariate Cox, HR = 0.06, 95% CI = 0.01–0.65, *p* = 0.02; Figure 3E), and the pooled datasets of CRC60 and CRC27 (multivariate Cox, HR = 0.46, 95% CI = 0.28–0.75, *p* = 2 × 10^−3^; Figure 3F). These results demonstrate that the 3-GPS exhibited robust performance in all the training and validation datasets.

Figure 4 depicts the 3-GPS score for each sample within the three datasets. It can be observed that, of the eleven non-responder samples of CRC47, six scored 0 and five scored 1. Among the 36 responder samples, 20 samples scored 3, 13 samples scored 2, 1 sample scored 1, and 2 samples scored 0. It can be observed that the three gene pairs constituting 3-GPS have a similar expression pattern in the majority of samples, but there are also instances where they differ. A similar phenomenon was observed in the samples from CRC60 and CRC27. Here, 24 samples were predicted to be non-responders in the CRC60, of which 17 scored 1 and 7 scored 0. Another 36 samples were predicted to be responders, of which 23 were scored 3 and 13 were scored 2. In CRC27, 15 samples were predicted to be non-responders, all of which scored 1. Additionally, twelve samples were identified as responders, comprising six samples with a score of 3 and six samples with a score of 2. In summary, the three gene pairs play different roles and cooperate to form the 3-GPS score in the samples.

### 2.3. Mechanism of the Signature Genes in 3-GPS

In order to elucidate the mechanism of 3-GPS genes in predicting the pathological response of FOLFIRI chemotherapy, a co-expression analysis was conducted. A total of 2652, 738, 5737, 654, 12, and 2 genes were found to be co-expressed with *IKBKB-DT*, *OR7E14P*, *PLXNA4*, *RABGAP1*, *CSTV,* and *LOC401052*, respectively (Spearman’s rank correlation, FDR < 0.01 and |r| > 0.5). A Kyoto Encyclopedia of Genes and Genomes (KEGG) enrichment analysis was performed on these genes. At an FDR threshold of 0.05, 10, 6, 1, 4, and 1 pathways were identified as enriched for the co-expressed genes of *IKBKB-DT, OR7E14P*, *PLXNA4*, *RABGAP1,* and *CTSV*, respectively (Figure 5A–E). The co-expressed genes of *IKBKB-DT* and *PLXNA4* were found to be enriched in the neuroactive ligand−receptor interaction pathway. Published research has demonstrated that the inhibition of the neuroactive ligand–receptor interaction pathway can enhance the efficacy of immunotherapy in colon cancer [14]. The co-expressed genes of *OR7E14P* and *RABGAP1* were found to be enriched in the cell cycle, oocyte meiosis, and progesterone-mediated oocyte maturation pathway.

The drug related genes of 5-FU, leucovorin, and irinotecan from DrugBank [15] were employed to construct a PPI network with the 3-GPS genes, based on the PPI information from the STRING database. Furthermore, the PPI network was extracted by identifying significantly correlated gene pairs (Figure 5F). The results demonstrated that the expression of signature genes *IKBKB-DT* and *PLXNA4* was both positively correlated with drug-related genes *SLC19A1*, *SLC22A6*, *SLC22A7,* and *ABCC11*. Additionally, *PLXNA4* exhibited positive co-expression with *ABCC3, TERT,* and *FOLR2* and negative co-expression with *ABCC4*, *PPAT*, *DHFR*, *SLC25A32,* and *GGH*. *PLXNA4* has been demonstrated to promote tumor progression and tumor angiogenesis by enhancing the signaling of VEGF and bFGF [16]. *RABGAP1* was found to be positively co-expressed with *ABCC3* (Figure 5F). The extensive interactions between these signature genes and drug-related genes suggest that these signature genes may play an important role in the FOLFIRI chemotherapy of mCRC.

### 2.4. Relationship Between the Signature Genes and Immune Cell

By employing the ESTIMATE algorithm, we were able to ascertain the estimate, immune and stromal scores, as well as the tumor purity levels, between the responder and non-responder groups. Our findings revealed that the responder group exhibited significantly elevated estimate, immune and stromal scores, in addition to significantly reduced tumor purity levels, when compared to the non-responder group (*t*-test, *p* < 0.05, Figure 6A). Subsequently, the extent of immune infiltration in 22 different kinds of immune cells in mCRC patients was assessed using the CIBERSORT (Figure 6B). Specifically, non-responders exhibited elevated levels of plasma cells and memory-resting CD4+ T cells (*t*-test, *p* < 0.05, Figure 6B). Plasma cells and CD4+ T cells have been linked to the prognosis of colorectal cancer, as evidenced by previous studies [17,18].

Further analysis revealed a significant negative correlation between the expression of signature gene *OR7E14P* and estimate, immune, and stromal scores (Pearson test, *p* < 0.05, Figure 6C). Additionally, its expression exhibited a positive correlation with the quantities of plasma cells and memory-resting CD4+ T cells (Pearson test, *p* < 0.05, Figure 6D). These findings suggest that the high expression of *OR7E14P* may suppress immune infiltration in mCRC and promote mCRC progression. Indeed, there was a positive correlation between *OR7E14P* expression and tumor lesion size (Spearman’s rank correlation, r = 0.45, *p* = 0.04). A negative correlation was observed between *PLXNA4* and the amount of memory-resting CD4+ T cells, while a positive correlation was evident between *PLXNA4* and regulatory T cells (Pearson test, *p* < 0.05, Figure 6D). *PLXNA4* plays a role in the trafficking and exclusion of cytotoxic T cells in cancerous tissue, as evidenced by research [19].

## 3. Discussion

In this study, a signature comprising three gene pairs (3-GPS) was developed with the objective of predicting the response of an mCRC patient to FOLFIRI chemotherapy. 3-GPS could discriminate the FOLFIRI responders and non-responders accurately. Moreover, the FOLFIRI responder groups predicted by 3-GPS had significantly better PFS than the predicted non-responder groups.

The treatment options for metastatic colorectal cancer are highly complex and are influenced by several factors, including the different molecular subtypes and primary tumor sites [20,21]. 5-FU is the backbone of mCRC treatment. The majority of first- and second-line clinical trials have investigated diverse combinations based on 5-FU [22]. In accordance with the ASCO (American Society of Clinical Oncology) Guideline, FOLFOX (5-FU, leucovorin, and oxaliplatin) and FOLFIRI are recommended as first-line therapy for patients with microsatellite stable (MSS) or proficient mismatch repair (pMMR) mCRC [3]. It is recommended that pembrolizumab should be offered as a first-line therapy to patients with microsatellite instability-high (MSI-H) or deficient mismatch repair (dMMR) mCRC. And, numerous 5-FU-based signatures have been developed to predict the response of CRC patients to diverse regimens incorporating 5-FU [23,24]. For example, Hou et al. [23] established a 5-FU-based prognostic signature for predicting the efficacy of chemotherapy and immunotherapy in colorectal cancer patients. Similarly, Song et al. [24] established 5-FU-based signatures for predicting chemotherapy response in right- and left-sided colon cancer.

However, CRC patients respond differently to 5-FU alone and 5-FU-based regimens. For example, studies show that a subgroup of colon cancer patients can respond better to FOLFIRI than to 5-FU alone [25]. Del Rio et al. have reported a gene expression signature in advanced CRC patients to select drugs and respond to the use of FOLFIRI [26]. However, this study used only 21 CRC samples, and validation datasets are lacking. In contrast, a total of 108 mCRC samples collected from four labs were used in this study. Moreover, not only was the response information used, but also the survival information was used to evaluate the signature. More importantly, compared to markers that are scored directly using gene expression values, our markers can achieve individualized drug response prediction without the need to use a batch of samples for data normalization.

Few studies have been conducted to investigate the role of the six genes that comprise the 3-GPS (*IKBKB-DT*, *OR7E14P*, *PLXNA4*, *LOC401052*, *RABGAP1*, and *CTSV*) in the context of chemotherapy for colorectal cancer. However, the results of protein–protein interaction and correlation analyses suggest that these genes have extensive interactions and expression correlation with drug-related genes, including 5-FU, leucovorin, and irinotecan. Consequently, the co-expressed genes of these genes were found to be enriched in pathways of particular importance to colorectal cancer immunotherapy. The analysis of immune infiltration revealed a significant correlation between the expression of these genes and the proportion of immune cells in colorectal cancer samples. These findings suggest that these genes may play an important role in the context of FOLFIRI therapy for colorectal cancer, potentially influencing immunomodulation in colorectal cancer.

It should be acknowledged that this study is not without some limitations. The sample size was limited due to the fact that we focused exclusively on the chemotherapy response to FOLFIRI in patients with mCRC. Accordingly, the statistical power of the statistical analyses conducted in this study was calculated. The power of Spearman’s rank correlation analysis, which was conducted using 21 samples to identify genes associated with response to chemotherapy, was 0.27. The efficacy is indeed poor. However, this effect is limited because only some, but not all, of the chemotherapy response genes are required to construct the signature. Moreover, the power of the Wilcoxon rank-sum test for identifying pairs of genes associated with response to chemotherapy using 47 CRC samples was 0.913. It is noteworthy that all survival analyses achieved a power of 0.8 or higher for the survival validation of the signature. It can therefore be concluded that the signature identified in this study is adaptable in colorectal cancer. It would undoubtedly be beneficial to expand the sample size, which would necessitate the collection of additional samples from high-volume hospitals [27]. Concurrently, the acquisition of additional samples with enhanced information content will facilitate more comprehensive analyses, including the impact of microsatellite instability, RAS gene mutation, and other factors on the signature. Consequently, the intention is to gather additional FOLFIRI chemotherapy samples from our affiliated hospitals to further substantiate the signature. Presently, the FOLFIRI plus immunotherapy regimen is being employed with increasing frequency. Based on the findings of this study, it is a promising avenue for further research to construct a predictive model of chemotherapy response to FOLFIRI plus immunotherapy and to investigate the underlying mechanisms.

## 4. Materials and Methods

### 4.1. Data Sources and Preprocessing

A total of 108 colorectal cancer samples that had undergone FOLFIRI chemotherapy were collected in this study. A summary of the clinical and pathological characteristics of these patients is provided in Table 2. Tumor response was evaluated in accordance with RECIST recommendations for the assessment of cancer treatment in solid tumors [28,29]. Previous studies have demonstrated that the prognoses of colorectal cancer samples with stable disease (SD) show no difference to partial response (PR) [30]. Accordingly, the present study defines complete response (CR) and PR as response categories, whereas progressive disease (PD) is defined as a non-response category. Progression-free survival (PFS) was defined as the time elapsed between the commencement of the initial treatment regimen and the occurrence of disease progression or death. Patients who were alive and had not experienced disease progression were considered to be censored at the end of the last follow-up period. Tumors that originated in the splenic flexure, descending colon, or sigmoid colon were classified as left-sided colon tumors, whereas those in the appendix, cecum, ascending colon, hepatic flexure, or transverse colon were classified as right-sided colon tumors [31]. A summary of the clinical characteristics for each dataset is provided in Table 2.

The GSE72970 dataset comprises both response and survival data. The GSE62080 dataset comprises data on the size of tumor lesions, as well as information on responses and survival. The datasets GSE39582 and GSE104645 comprise solely survival data. The four datasets were divided into two groups, designated the training and validation groups (Table 3). The training and validation groups each consist of two sub-groups. The initial training dataset, designated CRC21, includes all 21 patients with tumor lesion information from GSE62080. The second training dataset comprises 47 mCRC patients with response data only, including 10 patients from GSE62080 and 37 patients from GSE72970, collectively designated CRC47. The initial validation dataset comprises 60 samples with survival information from GSE72970, designated as CRC60. The second validation dataset, designated CRC27, encompasses 12 patients from GSE39582 and 15 patients from GSE104645 and comprises solely survival information.

For colorectal cancer samples detected by the Affymetrix platform, the original mRNA expression data (.cel file) were obtained and the RMA (robust multichip average) method was employed for background adjustment. For the data measured by the Agilent platform, the processed data were downloaded directly. For all the aforementioned samples, the corresponding platform file was used to map each probe to the Entrez gene ID. If multiple probes correspond to the same Entrez gene ID, we calculate the average of the expression values of these probes. If a probe corresponds to multiple Entrez gene IDs or does not correspond to any Entrez gene ID, the probe and its expression profile data were excluded from the analysis.

### 4.2. Construction and Screening of Gene Pairs

Spearman’s rank correlation coefficient was used to identify the genes associated with the pathological response to FOLFIRI chemotherapy in patients with colorectal cancer. A list of gene pairs was generated based on the above genes. The gene pairs were then screened using the Wilcoxon rank-sum test.

An expression difference (ED) value was calculated for each gene pair (denoted as Gene i and Gene j). Gene pairs with an ED value of 0 or less were excluded from further consideration. The ED values for gene pairs (*i*, *j*) are calculated as follows:$$ED_{ij}=\frac{1}{n1}\times \sum_{n=1}^{n1}\left(E_{ni}-E_{nj}\right)-\frac{1}{n2}\times \sum_{n=1}^{n2}\left(E_{ni}-E_{nj}\right)$$

In this context, *n*1 represents the number of responder samples, while *n*2 represents the number of non-responder samples. The term $E_{ni}$ represents the expression of gene i in sample *n*.

### 4.3. Calculating the F-Score of Each Gene Pair

The F-score for each gene pair was calculated using the following formula:$$F\text{-}\mathrm{score}=\frac{\mathrm{TP}+\mathrm{TN}}{\mathrm{TP}+\mathrm{FN}+\mathrm{TN}+\mathrm{FP}}$$
where true positive (TP) represents the number of samples correctly identified as responders and true negative (TN) represents the number of samples correctly identified as non-responders. Similarly, the terms false positive (FP) and false negative (FN) refer to the number of false positives and false negatives, respectively.

### 4.4. Developing the Predictive Gene Pair Signature for FOLFIRI Therapy

Firstly, response genes associated with the degree of response to FOLFIRI chemotherapy were identified by Spearman’s rank correlation. Subsequently, gene pairs comprising at least one response gene were selected based on the real ${\mathrm{ED}}_{ij}$ as the rank-selected gene pairs. The gene pairs with significantly increased ability to predict patient responses were designated as response-associated gene pairs by using the exact binomial test. All the response-associated gene pairs were considered as potential candidates for the development of a predictive gene pair signature (GPS). For each response-associated gene pair, the accuracy and F-score were calculated. A forward selection procedure was employed to identify the gene pairs with the largest F-score for predicting patient response. In summary, the gene pairs were arranged in a descending order according to their F-scores, with the top 20 gene pairs with the highest F-scores being selected as seeds. For each seed gene pair, an additional candidate gene pair was incorporated into the signature set with the objective of increasing the F-score until no further improvement could be achieved. The gene pairs with the highest F-scores, derived from the 20 seeds, were selected as the predictive signatures for FOLFIRI.

### 4.5. Area Under the Curve and Survival Analysis

The area under the receiver operating characteristic curve (AUC) was employed to evaluate the predictive performance of signatures. The calculation was performed in R (version 4.4.1) using “pROC” package (version 1.18.5) [32]. A univariate Cox proportional hazards regression model was employed to assess the association between the signature and patient survival. A multivariate Cox model was employed to assess the independent correlation between the signature and patient survival after adjustment for clinical variables such as stage, age, gender, and tumor location. The Kaplan–Meier method was employed to estimate survival curves [33].

### 4.6. Co-Expression Analysis and Human Protein–Protein Interaction Analysis

Spearman’s rank correlation was employed to identify the co-expression genes of the 3-GPS genes. The protein–protein interaction (PPI) data were downloaded from the STRING database (https://string-db.org/ (accessed on 12 September 2024)) [34]. Genes involved in the transport, metabolism, and other downstream effects of 5-FU, leucovorin, and irinotecan (referred to here as ‘drug related genes’) were sourced from the DRUGBANK database (https://go.drugbank.com/ (accessed on 10 September 2024)) [15]. Subsequently, the drug-related genes were employed to construct a direct PPI network with the 3-GPS genes, utilizing the aforementioned PPI data. Furthermore, the PPI network was extracted by identifying gene pairs with significant correlation.

### 4.7. Relationship Between Signature and Tumor Immune Microenvironment

The Estimation of Stromal and Immune cells in Malignant Tumor tissues using Expression (ESTIMATE, version1.0.13) [35] algorithm was employed to ascertain the stromal, immune, and estimate scores and tumor purity of each mCRC sample. The CIBERSORT approach (version 1.03) [36], which quantifies the relative abundance of immune cell types based on specific gene expression profiles, was used to assess the distribution (normalized to 1) of 22 immune cell types in mCRC patients. Furthermore, a two-sample *t*-test was employed to ascertain the discrepancy in immune cell composition between responders and non-responders with mCRC. The Pearson correlation coefficient was employed to investigate the potential correlation between the expression of the signature gene and immune cells in the context of mCRC.

### 4.8. Power Calculation

The power calculation for Spearman’s rank correlation was conducted in R using the “pwr” package (version 1.3-0) [37]. Similarly, the power calculation for the Wilcoxon rank-sum test was performed in R using the “wmwpow” package (version 0.1.3). The power calculation for survival analysis was performed in R using the “powerSurvEpi” package (version 0.1.3) [38].

## 5. Conclusions

In conclusion, the REO-based qualitative transcriptional signature has been demonstrated to be an effective method for identifying mCRC patients who may benefit from the FOLFIRI regimen. Furthermore, these signature genes may play an important role in the FOLFIRI chemotherapy of mCRC.

## Figures and Tables

**Figure 1 ijms-25-12771-f001:**
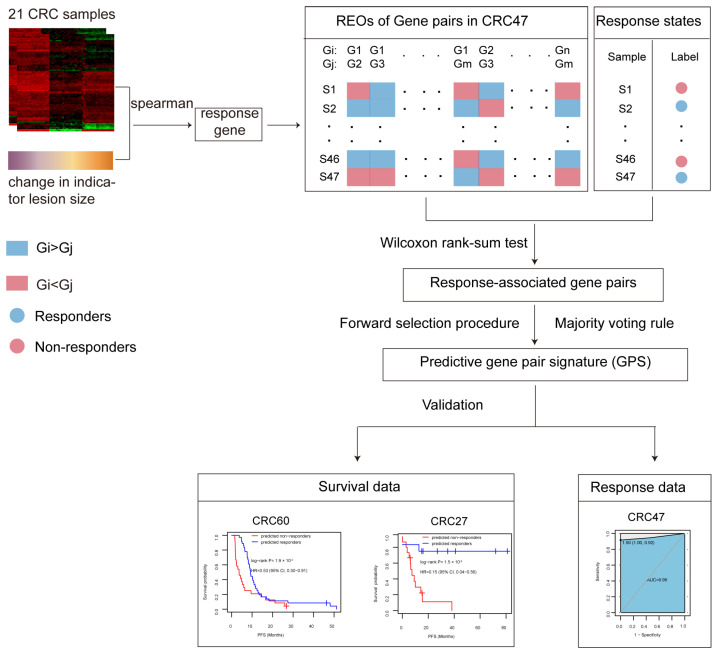
Flowchart for the identification and validation of the relative expression ordering (REO)-based signature in the present study.

**Figure 2 ijms-25-12771-f002:**
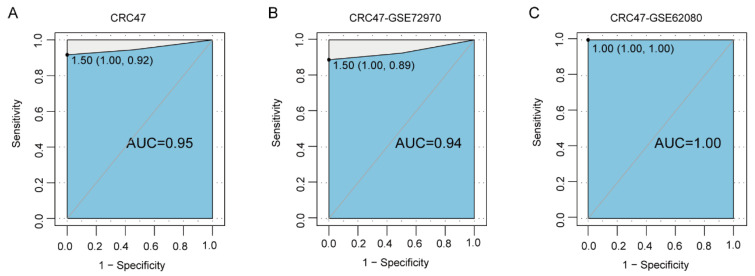
Performance of 3-GPS signatures in identifying responder patients with FOLFIRI treatment. (**A**–**C**): The AUC in training datasets CRC47, CRC47-GSE72970, and CRC47-GSE62080.

**Figure 3 ijms-25-12771-f003:**
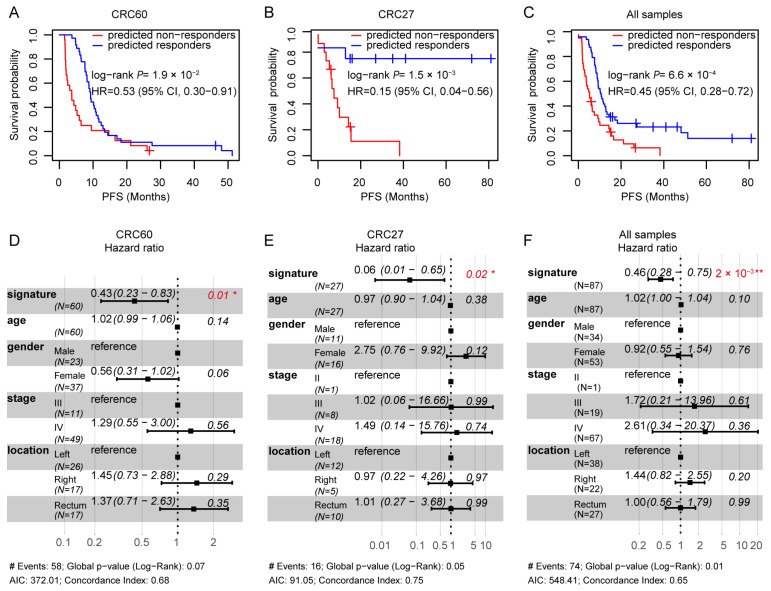
Univariate and multivariate Cox analysis for 3-GPS. (**A**–**C**): Univariate Cox analysis for 3-GPS in CRC60, CRC27, and the pooled datasets of CRC60 and CRC27. (**D**–**F**): Multivariate Cox analysis for 3-GPS in CRC60, CRC27, and the pooled datasets of CRC60 and CRC27 after adjusting for age, gender, stage, and tumor location. * *p*  <  0.05. ** *p*  <  0.01.

**Figure 4 ijms-25-12771-f004:**
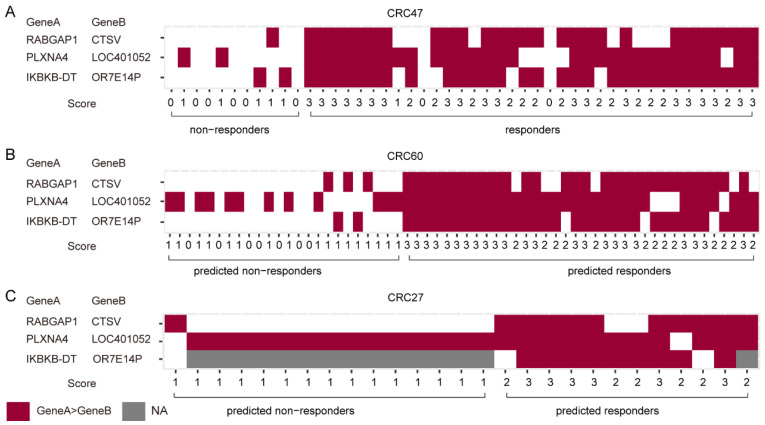
The distribution of 3-GPS scores in different datasets. (**A**–**C**): The scores distribution in CRC47, CRC60, and CRC27.

**Figure 5 ijms-25-12771-f005:**
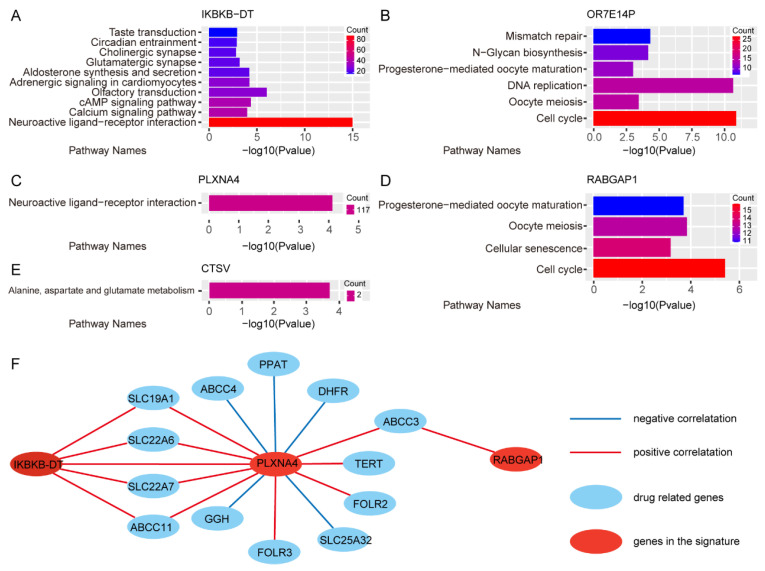
The function of genes in 3-GPS. (**A**–**E**) The enrichment result of co-expressed genes of *IKBKB-DT*, *OR7E14P*, *PLXNA4*, *RABGAP1,* and *CTSV*; (**F**) The co-expression network of genes in 3-GPS.

**Figure 6 ijms-25-12771-f006:**
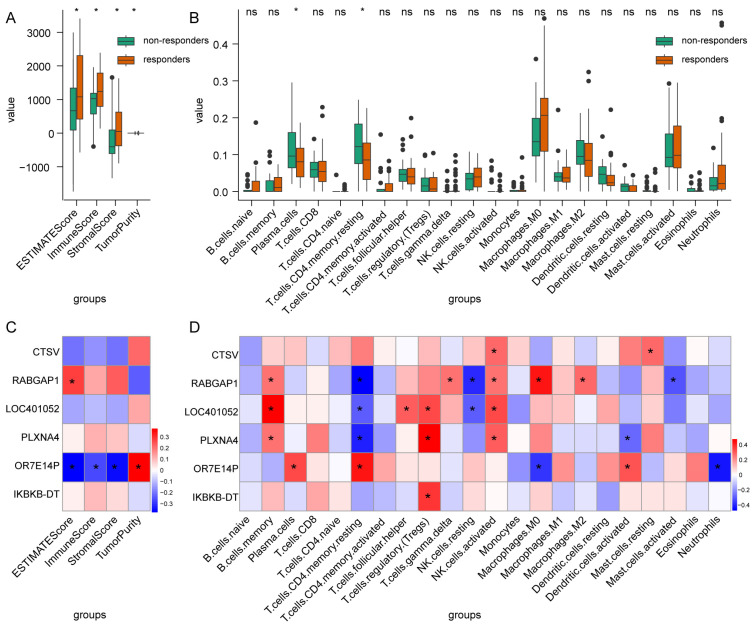
Immune infiltration in mCRC. (**A**): Comparison of the ESTIMATE score, immune score, stromal score, and tumor purity levels between responder and non-responder mCRC. (**B**): Comparison of the 22 different kinds of immune cell types between responder and non-responder mCRC. (**C**): The correlation between the expression of signature genes and ESTIMATE score, immune score, and stromal score in mCRC. (**D**): The correlation between the expression of signature genes and 22 different kinds of immune cell types in mCRC. * *p*  <  0.05, ns: no significant.

**Table 1 ijms-25-12771-t001:** The three gene pair in 3-GPS.

Gene Pairs	Gene i	Gene j
1	*IKBKB-DT*	*OR7E14P*
2	*PLXNA4*	*LOC401052*
3	*RABGAP1*	*CTSV*

**Table 2 ijms-25-12771-t002:** Baseline clinical and pathological characteristics of patients in this study.

Characteristic	GSE72970 (*n* = 60)	GSE62080 (*n* = 21)	GSE39582 (*n* = 12)	GSE104645 (*n* = 15)
Age (y)				
Mean ± SD	62.27 ± 10.87	59.10 ± 7.39	55 ± 10.11	69.33 ± 8.93
Median	62	60	54	68
Gender				
Male	37	11	7	9
Female	23	10	5	6
PFS (mo)				
Mean ± SD	10.70 ± 10.70	-	32 ± 30.73	10.22 ± 9.02
Median	8.07	-	21.5	7.47
PFS event				
0 (censored)	2	-	9	2
1	58	-	3	13
Response				
CR	7	-	-	-
PR	20	9	-	-
SD	23	11	-	-
PD	10	1	-	-
Location				
Right colon	17	3	7	5
Left colon	26	17	5	5
Rectum	17	1	-	5

Abbreviations: CR, complete response; PR, partial response; PD, progressive disease; SD, stable disease; PFS, progression-free survival, “-” means no such data.

**Table 3 ijms-25-12771-t003:** Training set and validation set used in this study.

Dataset	Data	Number	CR	PR	PD	Total	Platform
Training set	CRC21	GSE62080	-	-	-	21	GPL570 (Affymetrix)
	CRC47	GSE72970	7	20	10	37	GPL570 (Affymetrix)
		GSE62080	-	9	1	10	GPL570 (Affymetrix)
Validation set	CRC60	GSE72970	-	-	-	60	GPL570 (Affymetrix)
	CRC27	GSE39582	-	-	-	12	GPL570 (Affymetrix)
		GSE104645	-	-	-	15	GPL6480 (Agilent)

Abbreviations: CR, complete response; PR, partial response; PD, progressive disease; “-” means no such data.

## Data Availability

The datasets supporting the conclusions of this article are available in the GEO (https://www.ncbi.nlm.nih.gov/geo/ (accessed on 19 June 2020)).

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
