# Peer review of "Qualitative Transcriptional Signature for Predicting the Pathological Response of Colorectal Cancer to FOLFIRI Therapy"

_ijms, 2024, doi:10.3390/ijms252312771_

Round 1
Reviewer 1 Report
Comments and Suggestions for Authors
This paper describes the development of a gene signature to predict response to chemotherapy in colon cancer - this would be an important tool clinically preventing futile therapy for patients. The results are clearly presented - the figures are excellent. As the authors state a main limitation is sample size and in addition the number of statistical calculations made with such a small data set - while the stats section in the paper is very clear and the authors acknowledge the sample number, no reference is made to power calculation in the paper - this needs to be addressed. The discussion could be improved by greater exploration of other studies in this area of colon cancer therapeutics to contextualise the results - I would also like to see a discussion on future plans with this work to make it more robust with a larger data set
Reviewer 2 Report
Comments and Suggestions for Authors
Very interesting work that should be spread among colleagues who work routinely in the departments of genetics and medical oncology. In fact, the researchers who produced this paper asked themselves the question of why FOLFIRI, which is a mixture, if we can call it that, of drugs that is used in metastatic colon cancer in certain patients, has much less effect in others. Colleagues also provide us with numbers from international literature. They searched for the answer to their question in genetics. First, they used Spearman's rank correlation and the Wilcoxon rank-sum test to select the genes and gene pairs that responded to chemotherapy, respectively. Then, an optimization procedure was used to determine the final signature. The demonstrated impact on the immune system is very interesting. The methods explain in detail the procedure through which they managed to reach the conclusion of the study and for this reason it is absolutely reproducible. Obviously this is possible in high volume centers that therefore have the possibility to undertake this research (doi.org/10.3390/jcm12072708 to be cited in the bibliography). We agree with the weak points of the paper, limited number and response, it is surprising that patients with stable and unstable microsatellites have the same response. Excellent iconography that helps a lot in understanding the text good English, excellent bibliography
Round 2
Reviewer 1 Report
Comments and Suggestions for Authors
The authors have addressed my comments in my prior review